

# ELOVL2-AS1 inhibits migration of triple negative breast cancer

Mingda Zhu, Jingyang Zhang, Guangyu Li and Zhenzhen Liu

Department of Breast, Affiliated Cancer Hospital of Zhengzhou University & Henan Cancer Hospital, Zhengzhou, Henan, China

## ABSTRACT

In this study, we identified a key enhancer RNA (eRNA) region in breast cancer (BRCA) by applying an integrated analysis method. Reported eRNA region and genes affected by them were selected as presumed target pairs. Kaplan–Meier (KM) survival and correlation analyses were performed to screen valuable eRNA region. Based on the KM value and its correlation with the paired target genes, we carefully selected ELOVL2-AS1 as a potential key eRNA region in BRCA. Subsequently, we analyzed the expression of ELOVL2-AS1 and ELOVL2 in four BRCA subtypes and in different BRCA cell lines. The expression of ELOVL2-AS1 and ELOVL2 in triple negative breast cancer (TNBC) was significantly lower than those in Luminal A. After that, we analyzed the function of genes that are positively correlated with ELOVL2-AS1. We found that the co-expression gene mainly related to cilia and cilia characteristics of TNBC is significantly weaker than that of Luminal A. Considering the stronger invasion and metastasis of TNBC (compared with Luminal A) and the close relationship between decreased cilia and metastasis, we overexpressed ELOVL2-AS1 in TNBC and observed its effect on cell migration. The results show that it can inhibit the migration of TNBC. Finally, we analyzed the assay for transposase-accessible chromatin sequencing data, chromatin interaction analysis with paired-end tag sequencing data, and chromatin immunoprecipitation sequencing data and identified the chromatin interaction between ELOVL2-AS1 and ELOVL2, suggesting a direct regulatory interaction.

## INTRODUCTION

According to the latest data from the International Agency for Research on Cancer of the World Health Organization in 2020, breast cancer (BRCA) has the highest incidence rate among all cancer types. On the basis of clinicopathological criteria, BRCA can be divided into four types: (1) Luminal A, (2) Luminal B, (3) Erb-B2 overexpression, and (4) Basal-like (*Goldhirsch et al., 2011*). Although TNBC is not completely equivalent to Basal-like breast cancer, we generally treat them as a subtype (*Badve et al., 2011*). TNBC is the most intractable histopathological subtype of BRCA and is associated with high recurrence and metastasis rates (*Sharma, 2016*). Concurrently, TNBC is negative for estrogen receptor, progesterone receptor, and human epidermal growth factor receptor 2 (HER-2); hence, it lacks the targets for endocrine therapy and anti-HER-2 therapy. Previous studies have opined that there is no effective systemic treatment for TNBC except

Corresponding author
Zhenzhen Liu,
zlyyliuzhenzhen0800@zzu.edu.cn

chemotherapy. However, long-term chemotherapy will produce many toxic and side effects, such as diarrhea, mucositis, nausea and alopecia (*Kuchuk et al., 2013*). At the same time, not all TNBC patients are sensitive to chemotherapy. Therefore, it is essential to identify effective and characteristic biomarkers for targeted therapy.

*Jiang et al. (2019)* comprehensively analyzed the clinical, genomic, and transcriptomic data of a cohort of 465 primary TNBC patients in 2018. They further classified TNBC into four transcriptome-based subtypes: (1) luminal androgen receptor, (2) immunomodulatory, (3) basal-like immune-suppressed, (4) mesenchymal-like. Putative therapeutic targets or biomarkers have been identified for each subtype. However, it's still necessary to find therapeutic target from other biological perspectives. So far, although some studies have been carried out on enhancer RNAs (eRNA) and BRCA cell lines (*Li et al., 2013*; *Vucicevic et al., 2015*), there have been few investigations on the relationship between eRNA region and characteristics of breast cancer subtypes. In this scenario, we attempted to find key eRNA region related to BRCA subtypes from the list of eRNAs provided by previous studies. Such eRNA region and eRNAs are likely to serve as new therapeutic targets for TNBC.

## MATERIALS & METHODS

### Data extraction and eRNA analysis in BRCA

Gene expression and clinical data of BRCA were acquired from the University of California Santa Cruz Xena database (https://xena.ucsc.edu/). Raw data download link https://gdc-hub.s3.us-east-1.amazonaws.com/download/TCGA-BRCA.htseq_fpkm.tsv.gz; Full metadata https://gdc-hub.s3.us-east-1.amazonaws.com/download/TCGA-BRCA.survival.tsv; Full metadata. Firstly, we matched the BRCA clinical data and gene expression information. Thus, patients who have clinical and gene expression were involved in subsequent analysis. Meanwhile, we got a list concerning eRNAs and their target predicted using the PresSTIGE method (*Lam et al., 2013*; *Lee, Xiong & Li, 2020*). The relation between patients' OS and the level of putative eRNAs was identified by the R packages "survminer" and "survival". Co-expression analysis was carried out to evaluate the correlation between predicted targets and eRNAs. Putative region of eRNAs were candidates if they met the following standard: significant of OS (Kaplan–Meier $p < 0.05$) and co-expression with the predicted target ($r > 0.4$ and $p < 0.001$). Then, we selected an eRNA with low KM values (KM $<0.001$) and high correlation coefficients ($r > 0.9$) for further analysis. ATAC data obtain from the website (https://gdc.cancer.gov/about-data/publications/ATACseq-AWG). Transcriptome data are from TCGA database. The correlation analysis between peak and gene expression is done in R language (4.0.3). We downloaded the call sets from the ENCODE portal (https://www.encodeproject.org/) with the following identifiers: ENCFF328PXQ, ENCFF934LJL, ENCFF954BGW, ENCFF775RBW, ENCFF237WTX (*ENCODE Project Consortium, 2012*; *Davis et al., 2018*). ChIA-PET data were obtained from WASHU EpiGenome Browser In review (https://epigenomegateway.wustl.edu/) (*Zhou et al., 2013*). RNA-seq data of cancer cell lines were downloaded from the Cancer Cell Line Encyclopedia (https://portals.broadinstitute.org/ccle/about) (*Barretina et al., 2012*).

Metascape was used to visualized enrichment analysis result (http://metascape.org/) (*Zhou et al., 2019*). Protein interaction online analysis website STRING (https://string-db.org/) (*Szklarczyk et al., 2019*). GSE163214 was analyzed by GEO2R online software.

## Cell lines and cell culture

Human BC cell lines MDA-MB-231 was obtained from ATCC. Cells were maintained at 37 °C in a humidified cubicle containing 5% CO2 and 10% fetal bovine serum (FBS) in DMEM (Biological Industries Inc, Beit Haemek, Israel).

## Lentiviral construction of ELOVL2-AS1 overexpression

The sequences of human ELOVL2-AS1 were amplified and cloned into the pLV-007 lentivirus vector (BGI, China). Empty vector as a negative control (NC) were obtained from YBR (Shanghai, China). Constructs were verified by DNA sequencing. HEK-293T cells were transfected with plasmids by Lipofectamine$^{TM}$ 2000 (Invitrogen), envelope and packaging plasmids pMD2.G and psPAX2 (Addgene 3, Watertown, MA, USA). Viral particles were collected 48 h after infection. The MDA-MB-231 cell were infected by lentivirus (recombinant) in a 0.1% polybrene (Sigma-Aldrich) solution.

## Quantitative real-time polymerase chain reaction (qRT-PCR)

Total RNA of MDA-MB-231 was isolated by TRIZOL reagent (Solarbio, China) and reverse-transcribed into cDNA by a RT Reagent Kit (Vazyme, China). qRT-PCR using AceQ qPCR SYBR Green master Mix (Vazyme, China) was performed with an VIIA 7 (software version V1.3) real-time PCR system (Thermo Scientific) in triplicate, and the values were normalized to 18S (reference gene). The comparative CT (2- $\Delta\Delta$CT) method was applied to analyze the qRT-PCR data. The primer sequences used to quantify the target genes are: ELOVL2-AS1 primers (Sangon Biotech, China) 5′-AGAGAGCTGCCTTGCCCTTCC-3′ (sense) and 5′-AGAGTGGGTGTCTGGTGGTAAGC-3′ (antisense). ELOVL2 primers (Sangon Biotech, China) 5′-ATGTTTGGACCGCGAGATTCT-3′ (sense) and 5′-CCCAGCCATATTGAGAGCAGATA-3′ (antisense). 18SrRNA (Sangon Biotech, China) 5′-TTCCGATAACGAACGAGAC-3′ (sense) and 5′-GACATCTAAGGGCATCACAG-3′ (antisense). The experiment was repeated more than three times and included technical triplicates.

## Cell wound healing assay

Cells were cultured to full confluence in a 96-cell plate. Then, a scratch was generated using a cell scratch instrument. Cells were washed by phosphate-buffered saline (PBS) and cultured with serum-free medium. The scratches were then photographed at 0, 24, and 48 h, and cell migration was compared by measuring the gap size in each field. The experiment was repeated more than three times and included technical triplicates.

## Transwell migration assay

Chambers containing inserts with 8 μm pore were used in the experiment (Corning, NY, USA). Bottom chamber was used to hold DMEM with 30% FBS, while cells transfected with ELOVL2-AS1 OE, or NC in 100 μl serum-free medium was added to the Top chamber. The upper chamber was taken out after 24 h, then the bottom cartridge fixed by methanol

and crystal violet-stained to determine the migrated number of cells using a microscope to visualize the cells across randomly selected fields (Olympus, Tokyo, Japan). Cell numbers were quantified by Image J. The experiment was repeated more than three times and included technical triplicates.

## Cell proliferation assay

Six-well plates were used to house stable ELOVL2-AS1 OE and NC cells in colony forming experiments. The cells were maintained at 1,000 cells/well for 14 days or the number of cells in most single clones is greater than 50 in DMEM supplemented with 10% FBS, with media changed once every three days. Cells were then methanol-fixed and treated with crystal violet (Solarbio, China) prior to manual counting and photographing of the visible colonies. Colony number was measured using Image J software (vision 1.52a National Institutes of Health, USA). The experiment was repeated more than three times and included technical triplicates.

## Statistical analysis

Kaplan–Meier survival analysis was used to screen key eRNA region. Hypothesis testing method is log rank. It is a chi-square test with the degree of freedom being the number of time groups minus one. The exact $P$ values of the eRNA region screening are listed in the second column of Table S1. Spearman correlation analysis was used to analyze the correlation between eRNA region and potential target genes. The correlation coefficients and exact $P$ values between the eRNA region and potential target genes are column 4 and column 5. Spearman correlation analysis was also used to screen for genes that are positively correlated with ELOVL2-AS1 expression. The correlation coefficients and exact $P$ values of 429 genes positively correlated with the expression of ELOVL2-AS1 are the third and fourth columns of Table S2. In Fig. 1D, the unpaired $t$-test performed with GraphPad Prism 8 software. Figure 2 is to put 429 genes into the Metascape for Pathway and Process Enrichment Analysis. The statistical methods and related information are shown on the official website. Cilia-related differentially expressed genes (DEGs) in BRCA subtypes was conducted by R package limma ($|\log FC| > 0.5$, FDR $< 0.05$). The calculation of FDR adopts the Benjamini–Hochberg method.

# RESULTS

## Putative prognostic eRNAs in BRCA

1,082 BRCA patients were enrolled in the study after matching the patients' gene expression and clinical information. 2,695 eRNAs and 2,303 predicted target genes have been identified previously by the PreSTIGE algorithm (*Vucicevic et al., 2015*). We made use of these eRNA–target pairs to find out potential key eRNA region in BRCA. Finally, 35 target pairs were identified based on certain conditions (Kaplan–Meier [KM] log rank of $p < 0.05$ and correlation coefficient of $r > 0.4$ and $p < 0.001$) (Table S1). Then, we selected eRNA region with low KM values (KM $< 0.001$) and high correlation coefficients ($r > 0.9$) from Table S1 for follow-up studies. The flow chart of our work is presented in (Figure FC). Notably, because disease-free survival (DFS) is often used to describe prognosis in breast cancer

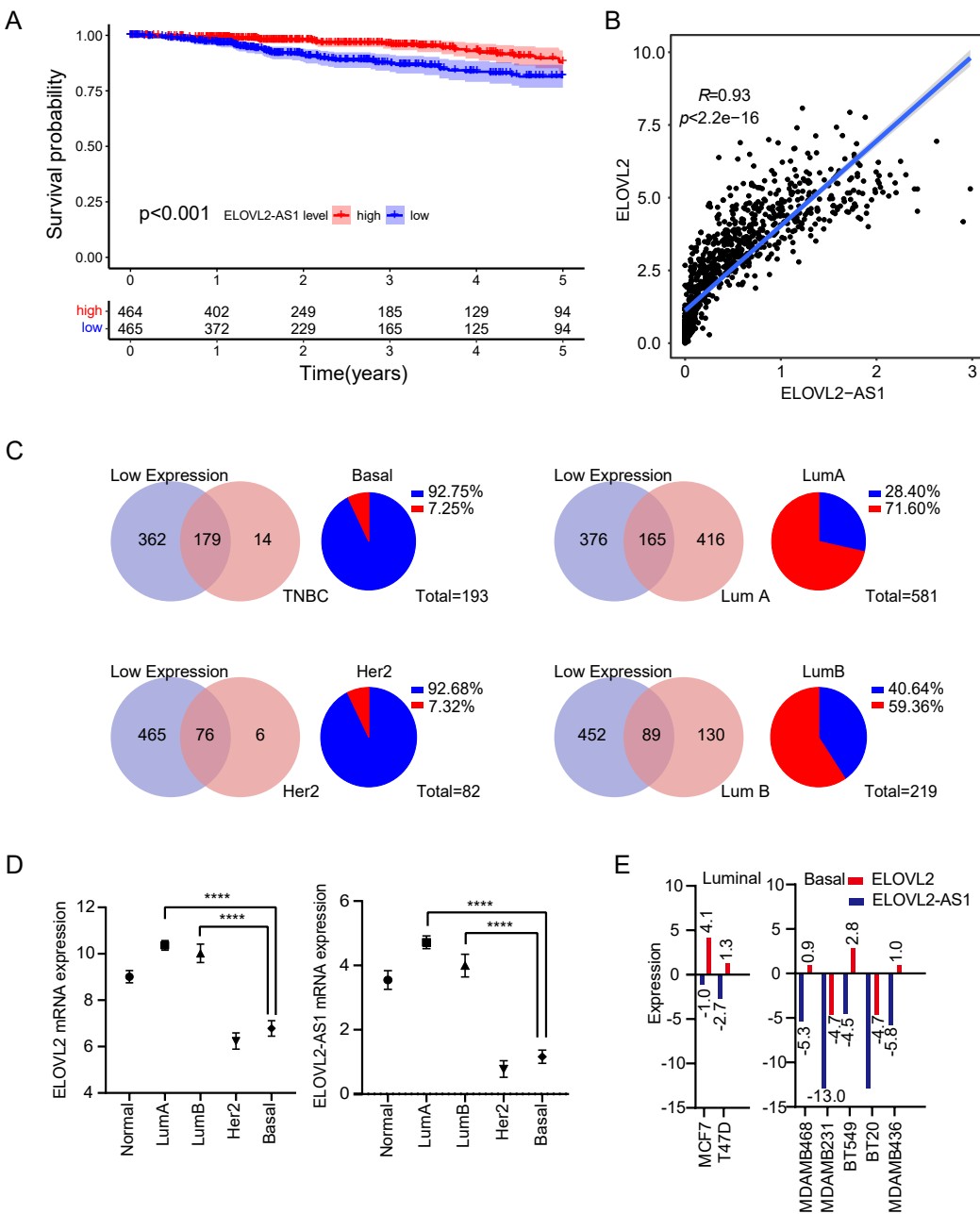

**Figure 1** **Expression analysis of ELOVL2-AS1 and ELOVL2 in breast cancer.** (A) Disease-free survival interval for ELOVL2-AS1. (B) The correlation between ELOVL2-AS1 and ELOVL2 expression (R = 0.93 $p$ < 2.2e−16). (C) Proportion of patients with low ELOVL2-AS1 expression in BRCA subtype in the TCGA. (D)The expression of ELOVL2-AS1 and ELOVL2 in the four subtypes. (E) The expression of ELOVL2 and ELOVL2-AS1 in different breast cancer cell lines.

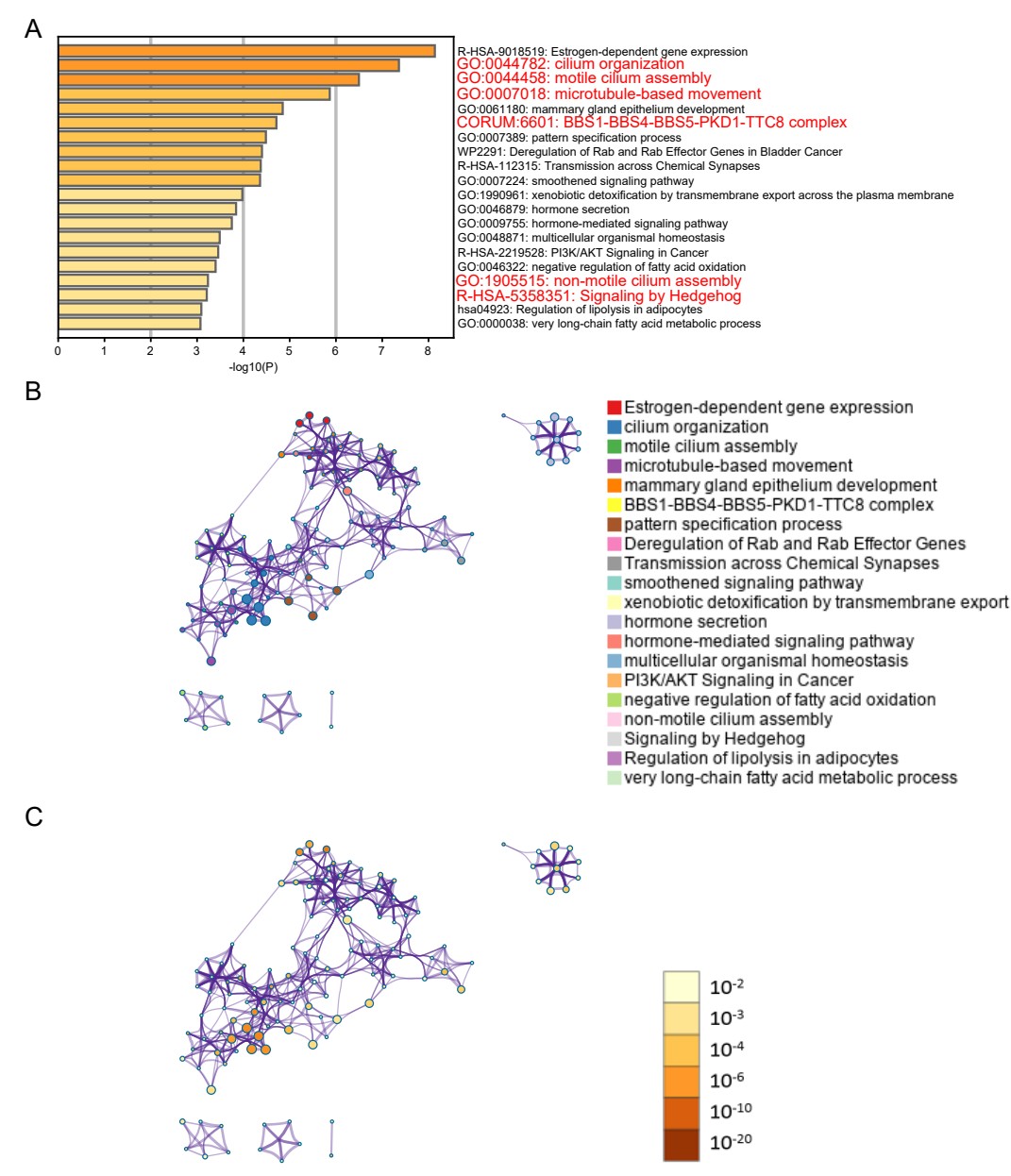

**Figure 2** **ELOVL2-AS1 related gene enrichment analysis.** (A) Bar graph of enriched genes (Top 20 clusters), colored by *p*-values. (B) (C) 367 of 429 genes can be identified in Metascape. Clustering network is visualized using Cytoscape5, where each node represents an enriched term and is colored first by its cluster ID and then by its *p*-value .

clinical work, we also analyzed disease-free interval (DFI) of ELOVL2-AS1 according to TCGA guidelines (Kaplan–Meier [KM] log rank of $p < 0.001$) (Fig. 1A), which is consistent with the results of overall survival (OS) (Fig. S1).

## Analysis of BRCA subtypes of ELOVL2-AS1 and ELOVL2

We observed that the lower the expression of elongation of very long chain fatty acids like 2 antisense RNA 1 (ELOVL2-AS1), the worse the prognosis (Fig. 1A). We were then curious about the subtype constitution of 541 BRCA patients; hence, we used the R package (TCGAbiolinks) to obtain the sample identities of the four BRCA subtypes and compared them with the 541 patients' identities. We were surprised that 92.75% of TNBC and 92.68% of HER2 positive BRCAs in the TCGA BRCA data involved low expression of ELOVL2-AS1. However, the proportion of Luminal A BRCA was only 28.4% (Fig. 1C). Then, we put forward a reasonable hypothesis: Is there a difference in the expression of ELOVL2-AS1 among the four BRCA subtypes? Based on this assumption, we analyzed the differences in gene expression of ELOVL2-AS1 and elongation of very-long-chain fatty acids-like 2 (ELOVL2) in normal breast tissue and four BRCA subtypes. As shown in Fig. 1D, the expression levels of ELOVL2 and ELOVL2-AS1 in the Basal and HER2 subtypes were significantly lower than those in the Luminal A and Luminal B groups and also lower than that in the normal group (95% CI [$-3.995$ to $-3.164$], $p < 0.0001$). So, ELOVL2-AS1 and ELOVL2 can be used as a biomarker to discern the subtype of BRCA.

Because the gene expression data of TCGA comes from tumor tissues, in order to exclude the influence of non-primary tumor cells in the tumor on gene expression, we obtained gene expression data of 2 Luminal cell lines and 5 Basal cell lines from the CCLE database. The expression of ELOVL2-AS1 and ELOVL2 in different cell lines showed that the expression in the Basal subtype was generally lower when compared with the Luminal subtype (Fig. 1E).

## ELOVL2-AS1 related gene prediction and enrichment analysis

It is generally believed that co-expressed gene sets participate in the same pathway or are subject to the same regulation or have similar biological functions. Using expression profile data to find mRNA co-expressed with lncRNA to predict the function of lncRNA is a commonly used strategy in bio-information analysis (Zhao et al., 2015). After clarifying the differences in the expression of ELOVL2-AS1 in the BRCA subtypes, we used mRNA data from the TCGA to perform co-expression analysis of ELOVL2-AS1. Through this analysis, we deciphered 429 genes are positively correlated with the expression of ELOVL2-AS1. The prediction screening criteria were correlation coefficient >0.4 and $p$-value < 0.001 (Table S2). Then, we conducted Gene Ontology (GO) and Kyoto Encyclopedia of Genes and Genomes (KEGG) enrichment analysis on 429 genes by R packages clusterProfiler (Fig. S2) (Yu et al., 2012). Enrichment analysis indicated that genes positively related to ELOVL2-AS1 are closely related to the cell structure of the cilia (Fig. 2A). Metascape was used to visualized enrichment analysis of 429 related genes. Moreover, pathway and process enrichment analyses were performed, with the network of the enriched terms. The enriched clusters for 429 related genes included those of "cilium organization," "motile

cilium assembly," "BBS1-BBS4-BBS5-PKD-TTC8 complex," "PI3K/AKT Signaling in Cancer," "Signaling by Hedgehog," and others (Fig. 2A). Figures 2B and 2C are network of enriched terms: 2B colored by cluster ID (nodes that own the same cluster ID are close to each other), 2C colored by *p*-value (terms including more genes tend to have a more significant *p*-value).

## Cilia-related differentially expressed genes (DEGs) and protein–protein interaction network (PPI) in BRCA subtypes

In the previous analysis, we found that the expression of ELOVL2-AS1 in Basal and HER2 subtypes was significantly lower than Luminal A and Luminal B. Therefore, genes that are positively linearly related to ELOVL2-AS1 should also have similar characteristics. Through GO and KEGG analyses, we found that the co-expressed gene group of ELOVL2-AS1 is linked to cell cilia. Consequently, we screened the 36 cilia-related genes from the GO analysis items for further examination. First, we obtained the expression data of the genes from the TCGA BRCA gene expression matrix and divided them into four BRCA subtype expression matrices (136 normal, 553 Luminal A, 200 Luminal B, 79 Her2, and 174 Basal) for DEGs analysis. Then, we used the protein interaction analysis website STRING to analyze the protein interactions of the genes and visualized the PPI network with CYTOSCAPE software. Between the differential gene analysis heatmap is the PPI analysis of the corresponding subtypes. We only discovered that 21 of the 36 cilia-related genes have protein interactions. The results assert that the differential expression of the 36 cilia-related genes in the four BRCA subtypes is different (Figs. 3A–3D). It is noteworthy that MAPT (log FC = −2.11, FDR = 6.70e−31, Not in PPI) and TTC36 (log FC = −2.16, FDR = 2.86e−42, Not in PPI) are down-regulated in Basal and up-regulated in Luminal A (compared to Normal respectively). CCDC170 (log FC = −2.34, FDR = 3.08e−56, Not in PPI) and DNALI1 (log FC = −2.66, FDR = 7.41e−50) are down-regulated in Basal and up-regulated in Luminal B (compared to Normal respectively).

Primary cilia are microtubule-based organelles that play important roles in cancer (*Higgins, Obaidi & McMorrow, 2019*). MAPT denotes the microtubule-associated protein tau, which is involved in the formation of the cilia. Therefore, the down-regulation of MAPT may affect the formation and functioning of cilia and indirectly affect cilia-related signaling pathways. Related research has shown that knocking down MAPT can promote the proliferation and invasion of renal clear cell carcinoma (*Han et al., 2020*). In addition, MAPT can alter the characteristics of the glioma mesenchyme by inhibiting EGFR-mediated NF-$\kappa$B and TAZ signaling pathways, inhibit the transformation of tumor cells into pericytes, normalize tumor blood vessels, and reduce the invasiveness of gliomas (*Gargini et al., 2020*). Overexpression of CCDC170 can significantly inhibit BRCA cell proliferation (*Wang et al., 2020*). Some scientists observed a statistically decrease in the proportion of ciliated cells on premalignant lesions and in invasive cancers (*Menzl et al., 2014*).

The above analysis proves that some co-expressed genes of ELOVL2-AS1 are also down regulated in TNBC. Meanwhile, the decreased expression of cilia genes associated with

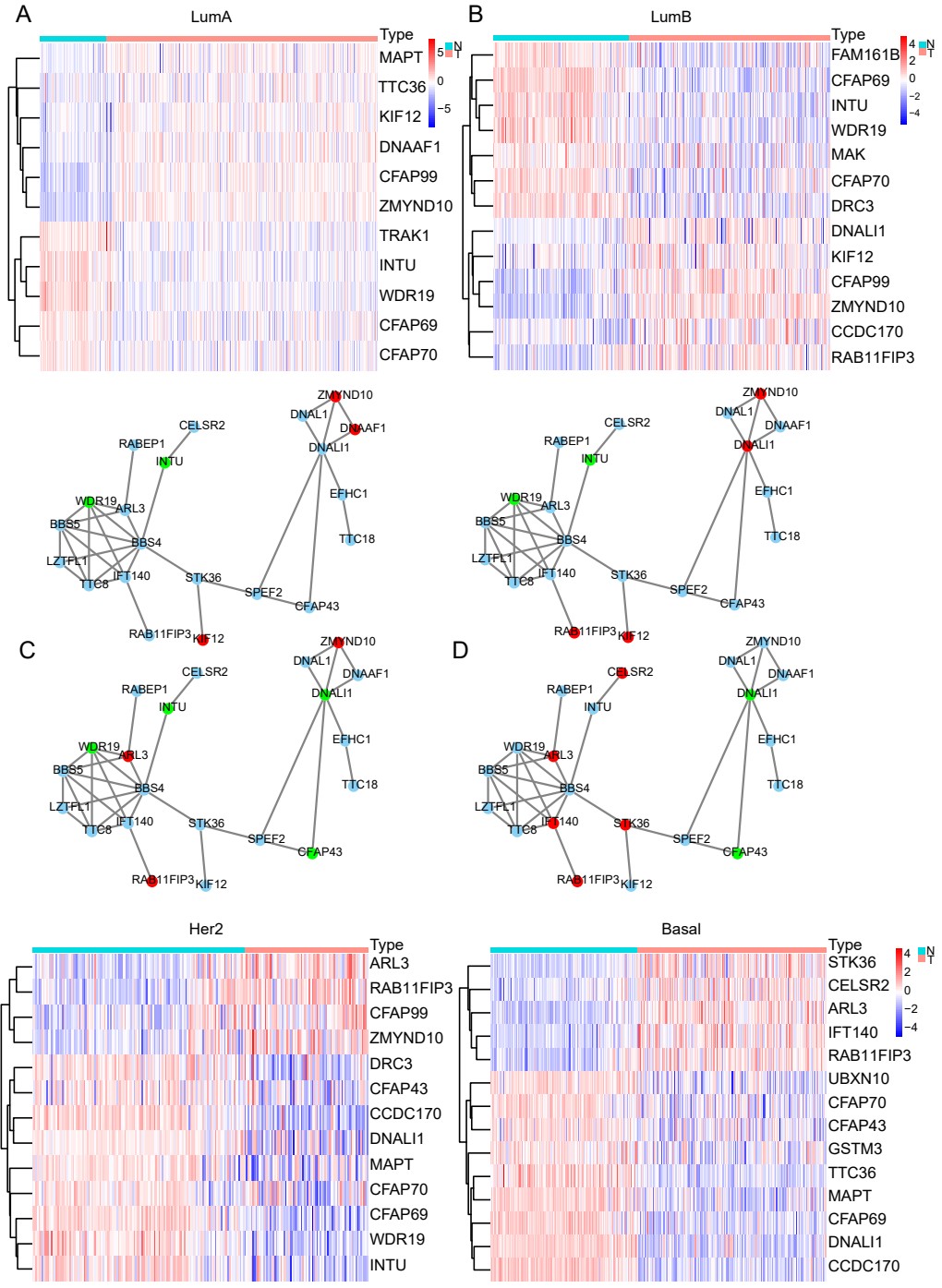

**Figure 3** **Cilia-related DEGs in BRCA subtypes.** (A–D) Heatmap of DEGs and PPI in Luminal A, Luminal B, Her2, TNBC. In the PPI red represents genes that are up-regulated in the subtype, and green represents the ones down-regulated in the subtype.

the migration or proliferation of tumor. However, it is not clear whether there is a direct regulatory relationship between these genes and ELOVL2-AS1.

## Cilia characteristics of the BRCA subtypes

After discovering differences in the expression of some cilia genes in breast cancer subtypes, in order to further evaluate the cilia characteristics of different subtypes, we conducted a co-expression analysis of 36 cilia genes (Figs. 4A–4E). The co-expression analysis of the 36 cilia-related genes in normal breast tissue and four BRCA subtypes indicated that the relationship among the cilia genes in the Basal-like (Fig. 4E) was significantly lower than that in the normal breast tissues (Fig. 4A) and Luminal A (Fig. 4B). This result suggests that the cilia properties of Basal-like BRCA cells are weakened when compared to normal breast tissues and Luminal A.

To further verify that the cilia properties of TNBC are weaker than those of Luminal A, we performed Gene Set Enrichment Analysis (GSEA) of Luminal A and TNBC (Fig. 5A). The 34 cilia-related enrichment analysis items were obtained from the GSEA official website, and the expression data of Luminal A and TNBC were extracted from the TCGA BRCA gene expression matrix. We then created a remark file (Luminal A 553 VS TNBC 174) for the analysis. GSEA analysis results showed that 20 gene sets are upregulated in Luminal A and 17 gene sets are significant at FDR < 25%. However, none of the gene sets are enriched in the phenotype TNBC. The remaining eight results of GSEA are given in (Fig. S3).

The weakening of the link among the cilia-related genes are associated with disorders of cilia formation. Many studies have shown that abnormal cilia formation is closely related to the abnormal Hedgehog signaling pathway (*Bangs & Anderson, 2017*). A significant relationship is considered to exist between the primary cilium and cancer via this pathway (*Goetz, Ocbina & Anderson, 2009*). Numerous cancer types, including lung cancer, basal cell carcinoma, prostate, and BRCA, have been proven to involve abnormal activation of the Hedgehog pathway (*Hassounah, Bunch & McDermott, 2012*). Some studies have alluded that the Hedgehog signaling pathway is abnormally activated in TNBC and that sonic Hedgehog regulates angiogenesis in TNBC and promotes the growth and metastasis of TNBC (*Jeng, Chang & Lin, 2020*). Mouse models also demonstrate that loss of cilia can increase tumor incidence in basal cell carcinoma (*Wong et al., 2009*). Abnormal expression of key genes in the Hedgehog signaling pathway shown in Fig. 5B. Based on the above analysis, we opine that the weakening of the characteristics of TNBC cilia may be related to the clinical features of easy metastasis (compared to Luminal A).

## ELOVL2-AS1 overexpression can affect ELOVL2 expression and BRCA cell migration

In the above analysis, we found several interesting things. First, ELOVL2-AS1 is significantly down-regulated in TNBC. Second, the decreased expression of certain co-expressed genes of ELOVL2-AS1 affects tumor proliferation and migration. Third, TNBC is more easily metastasized than Luminal A. Fourth, TNBC has obvious abnormalities in cilia and Hedgehog signaling pathways. Fifth, cilia abnormalities are closely related to tumor

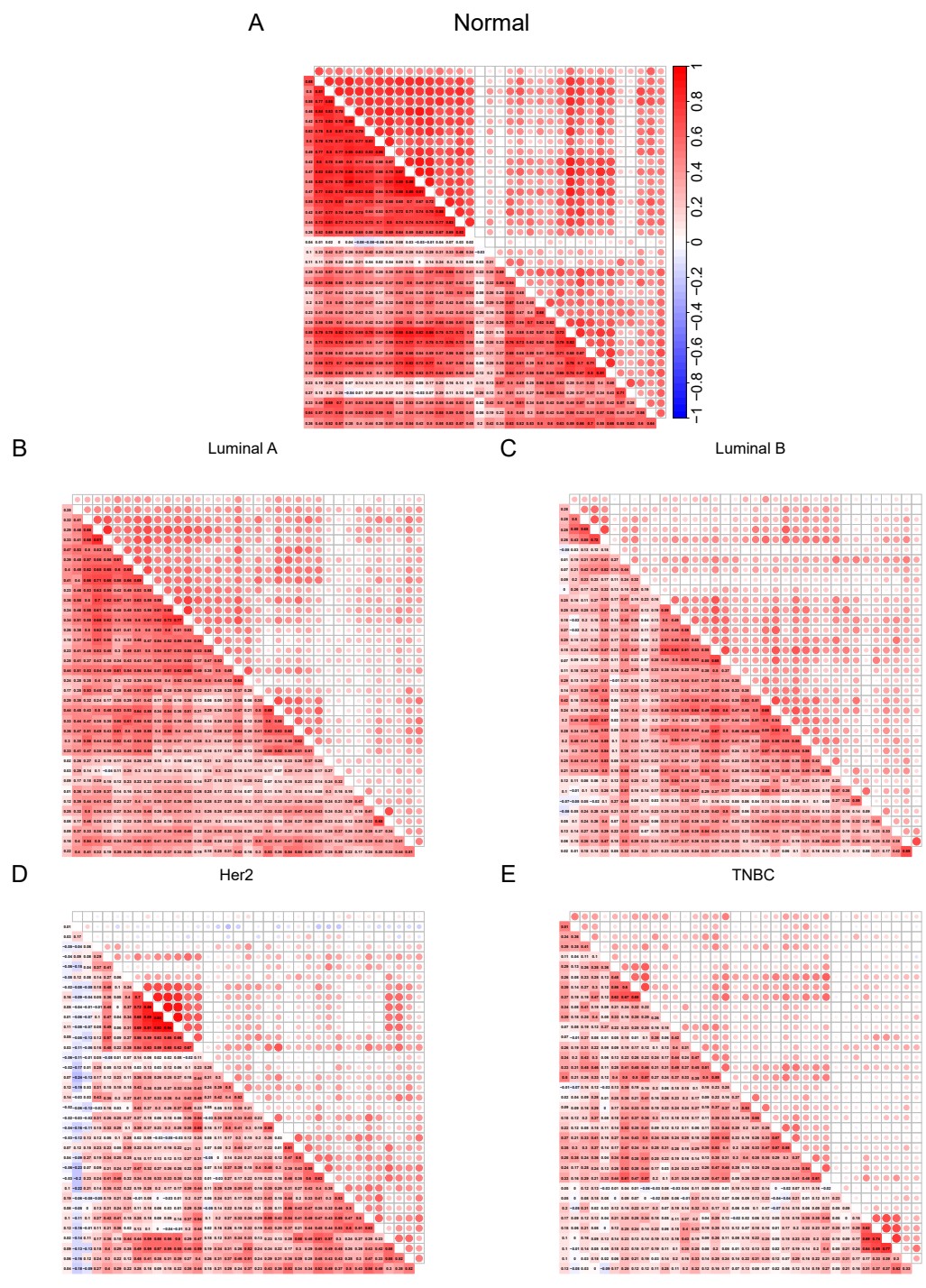

**Figure 4** **(A–E) Cilia characteristics of the BRCA subtypes.** Red represents positive correlation and blue represents negative correlation. The darker the square color in the lower left corner of each picture represents the greater the correlation, and the larger the circle in the upper right corner represents the greater the correlation.

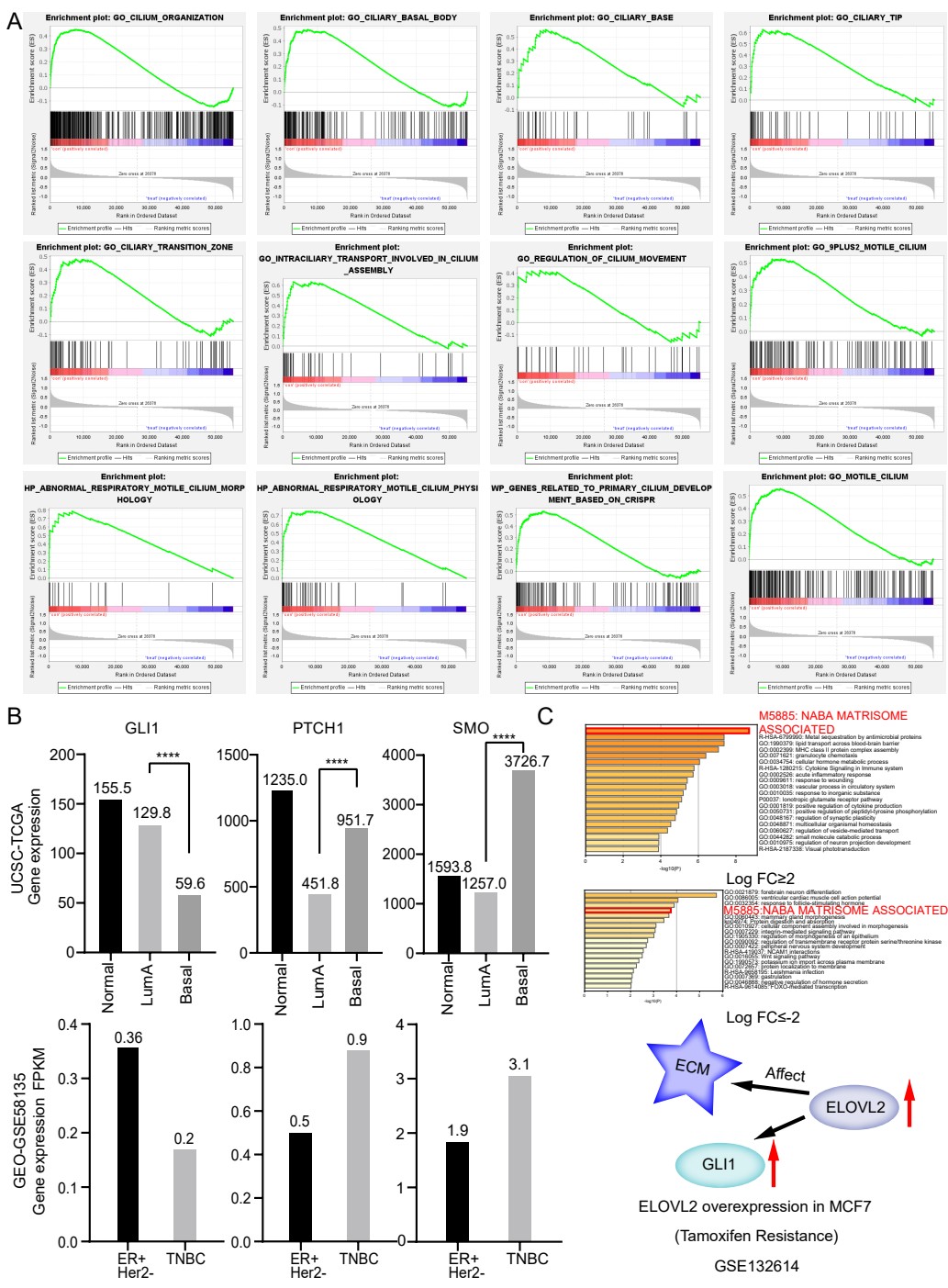

**Figure 5** **Ciliary characteristics and signal pathway analysis of luminal A and TNBC.** (A) GSEA between Luminal A and TNBC concerning cilia characteristics. (B) Abnormal expression of key genes in the Hedgehog signaling pathway. (C) ELOVL2 overexpression data analysis.

invasion and metastasis. Therefore, we speculate that the decreased expression of ELOVL2-AS1 may affect the migration of TNBC. Then, we constructed a lentiviral expression vector to overexpress ELOVL2-AS1 in MDA-MB-231. Transwell assay and Wound healing assay were used to examine the effect of overexpression on migration. The experimental results are shown in Figs. 6B and 6C ($t$ test, 24 h $p = 0.0013$, 48 h $p = 0.0004$). Overexpression significantly reduced the migration ability of 231 cells while it didn't have an impact on proliferation (Fig. 6D) ($t$ test, $p = 0.7049$, 95% CI [$-9.706$–$13.04$]). Unfortunately, it is still unclear whether ELOVL2-AS1 and cilia genes have a direct regulatory effect and thus affect the function of cilia.

## Analysis of possible regulatory relationships of ELOVL2-AS1 and ELOVL2

Previous research shows that ELOVL2 can catalyzes the synthesis of polyunsaturated very long chain fatty acid (C20- and C22-PUFA) (*Kitazawa et al., 2009*). Recent research shows that n-3 and n-6 PUFA selectively induced ferroptosis in cancer cells under ambient acidosis (*Dierge et al., 2021*). Therefore, the reduction of PUFA production caused by the reduction of ELOVL2 expression may lead to the imbalance of ferroptosis of tumors, which is beneficial to the occurrence and development of tumors. *Kang et al. (2019)* noted that the knockdown of ELOVL2 promotes the proliferation and migration of BRCA cells. Given that ELOVL2 may play an important role in tumorigenesis and development and the strong correlation between ELOVL2-AS1 and ELOVL2, we further analyzed the possible regulatory relationship between them (Fig. 1B) to explore whether AS1 can be used as a new therapeutic target.

First, we analyzed the Assay for Transposase-Accessible Chromatin sequencing data of BRCA and found that there are three peaks on chromosome 6 related to the expression of ELOVL2, namely peak1 (11043166–11043665), peak2 (11044316–11044815), and peak3 (11045321–11045820) (Fig. 7A). Next, we searched the gene structure of ELOVL2 and ELOVL2-AS1 on the genome browser ENSEMBL (http://asia.ensembl.org/index.html). Interestingly, the promoter range of ELOVL2 is 11042601–11045200 (hg37). The above analysis shows that PEAK1 and PEAK2 are located in the promoter region of ELOVL2. Then, we compared the relationship among the predicted eRNA region (ELOVL2-AS1, ENST00000456616), ELOVL2's promoter, and peak2. We found that overlapping regions exist among the three, which suggests that ELOVL2-AS1 may act on the promoter area and affect the expression of ELOVL2 (act along with transcription factor). This may be one of the mechanisms by which ELOVL2-AS1 affects the expression of ELOVL2.

In addition, relevant studies reported that eRNA can be combined with other protein factors to promote the formation of promoter–enhancer loops and affect gene expression (*Lee, Xiong & Li, 2020*; *Li et al., 2013*). Besides, there are three models to explain how eRNA affects gene transcriptional regulation (*Bose et al., 2017*; *Rahnamoun et al., 2018*; *Schaukowitch et al., 2014*; *Seila et al., 2008*; *Shi et al., 2018*; *Shii et al., 2017*; *Zhao et al., 2016*). Furthermore, we analyzed the chromatin immunoprecipitation sequencing (ChIP-Seq) and chromatin interaction analysis with paired-end tag (ChIA-PET) data near ELOVL2-AS1 and ELOVL2. This region habour classical enhancer features (enrichment

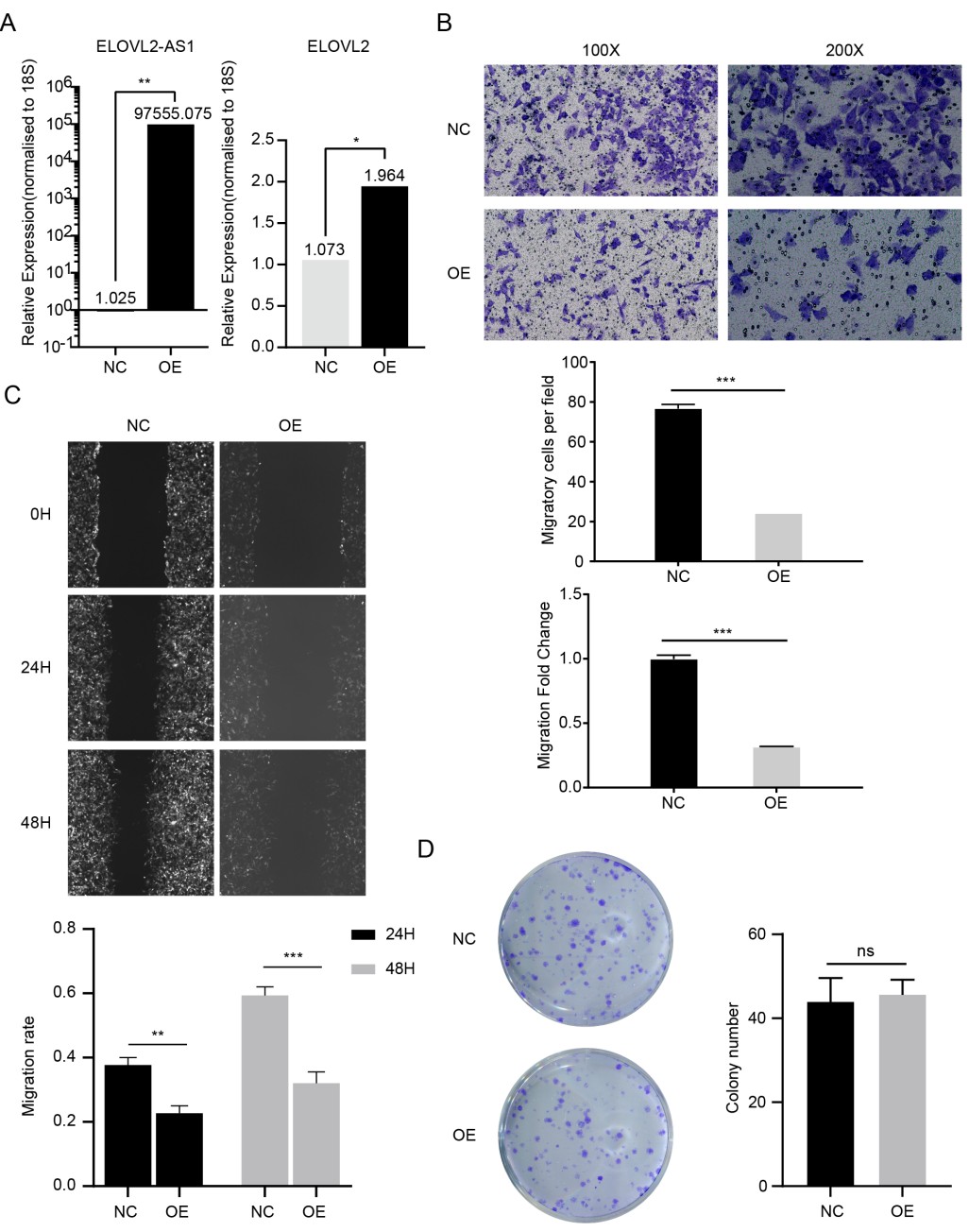

**Figure 6** **ELOVL2-AS1 overexpression can affect ELOVL2 expression and BRCA cell migration.** (A) qRT-PCR assay was performed to validate candidate genes regulated by ELOVL2-AS1. (B) (C) Transwell assay and Wound healing assay were used to examine the effect of overexpression on migration. (D) Cell proliferation assay was conducted to identify the proliferation change. All experiments were performed in triplicate and repeated at least three times. Detailed original pictures and data analysis files are stored in the following links: https://figshare.com/s/969e78d914fc0fadf3d4.

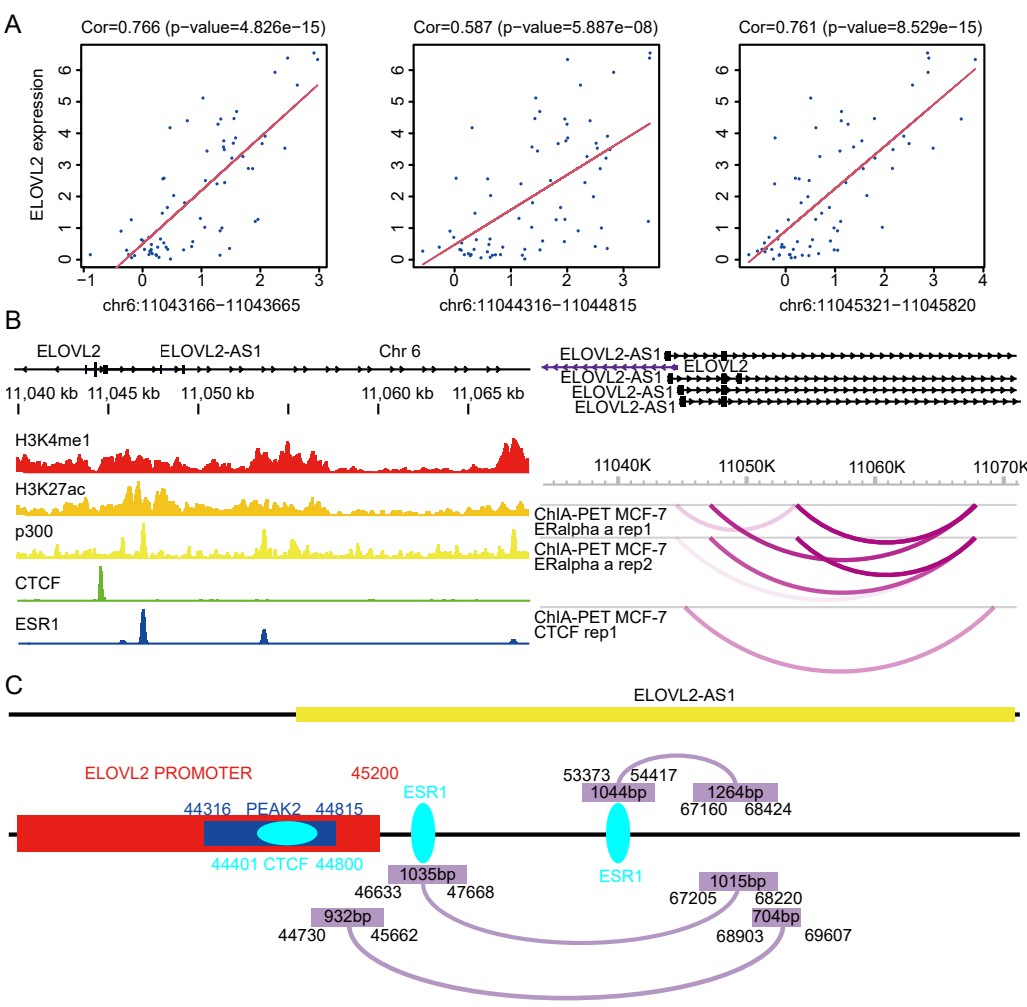

**Figure 7** **Possible regulatory relationships of ELOVL2-AS1 and ELOVL2.** (A) Correlation analysis between ATAC at specific sites of breast cancer and ELOVL2 expression. (B) Epigenetic features and chromatin interaction analysis of ELOVL2-AS1 and ELOVL2 region in breast cancer cell line. (C) Schematic diagram of possible regulatory relationships of ELOVL2-AS1 and ELOVL2.

of histone H3K4me1 modification). At the same time, the region exhibits active enhancer markers (strong enrichment of histone H3K27ac modification and binding of transcription co-factor p300) (Fig. 7B). CTCF ChIP-Seq data showed that there is a peak in 11044000–11045000. This is also the region in which the eRNA and PEAK2 are located, which indicates that the predicted eRNA-1 (704bp) may participate in the process of transcription regulation through the transcription factor CTCF. Besides, we observed chromatin interaction between ELOVL2-AS1 and ELOVL2 based on ESR1 ChIA-PET data, suggesting direct interaction (Fig. 7B). A study has demonstrated that estrogen can strongly stimulate the expression of ELOVL2 and that estrogen response elements (EREs) are found in the region near the promoter of ELOVL2 (the direction of the 5' end of the transcription start

site: −1274 and −2772). This finding indicates that eRNA-2 (1,015 bp) is involved in the transcriptional regulation of ELOVL2 by estrogen, estrogen receptor alpha, and ERE.

Although the PCR results show that the mRNA of ELOVL2 will also increase to varying degrees while overexpressing ELOVL2-AS1, the specific mechanism still needs more experimental verification (Fig. 6A) ($t$ test, $p = 0.00157$).

## DISCUSSION

In this study, we employed a series of bioinformatics methods to identify a key eRNA region in BRCA. Owing to differences in the expression of ELOVL2-AS1 in different BRCA subtypes, ELOVL2-AS1 can be used as a biomarker to distinguish the different subtypes of BRCA. Moreover, through GO and KEGG analysis of gene groups linearly related to ELOVL2-AS1 expression, we found that ELOVL2-AS1 is closely related to the function of cell cilia. Subsequently, we uncovered that the cilia properties of all types of BRCA are abnormal when compared to normal tissues according to the correlation analysis of cilia genes. The cilia characteristics of TNBC are more abnormal than the Luminal A. Simultaneously, the expressions of some important genes (GLI1, SMO, PTCH1) in the cilia-related Hedgehog signaling pathways exhibit a significant difference. Considering that previous studies have demonstrated that abnormal Hedgehog signaling pathways lead to increased TNBC metastasis, we proposed the hypothesis that AS1, which is closely related to cilia and Hedgehog signaling pathways, may affect the migration ability of TNBC and proved through cell experiments that AS1 can indeed inhibit TNBC migration. Finally, we found that ELOVL2-AS1 is very likely to act on ELOVL2 to affect the metastasis of BRCA cells by three ways (itself, eRNA-1 and eRNA-2) based on ChIP-seq and ChIA-PET data (Fig. 7B).

However, there are still some issues that need to be resolved: (1) How does ELOVL2-AS1 interact with CTCF and ERE directly as eRNA? (2) At present, it is only known that there is a linear correlation between ELOVL2-AS1 and the cilia-related genes MAPT ($r = 0.577$) and CCDC170 ($r = 0.484$), and it is not clear whether there is a direct regulatory relationship between ELOVL2-AS1 and the cilia-related genes MAPT and CCDC170. (3) Whether ELOVL2 has a direct or indirect relationship with cilia is not clearly understood. It has been proven that ELOVL2 is a type of elongation of very long chain fatty acids protein. It can catalyze the first and rate-limiting step of the long-chain fatty acid elongation cycle. This process allows the addition of two carbons to the chain of long- and very long-chain fatty acids (VLCFAs) per cycle. ELOVL2 may be involved in the formation of polyunsaturated VLCFAs that are enrolled in many biological processes (as precursors of membrane lipids and mediators) (*De Antueno et al., 2001*; *Kitazawa et al., 2009*; *Ohno et al., 2010*)). Because the cilia are located on the surface of the cell membrane, their formation and alteration are linked to the composition of the cell membrane (*Phua et al., 2017*). ELOVL2 may indirectly affect the production and functioning of cilia by influencing the synthesis of fatty acids and causing changes in cell membrane components.

Besides, *Zhang et al. (2020)* found that ELOVL2-AS1 is a potential diagnostic and prognostic marker for tamoxifen resistance. *Xiao et al. (2021)* observed that ELOVL2-AS1

is an immune-related long non-coding RNA that could be used to predict the survival and tumor microenvironment characteristics in BRCA. Sun et al. discovered ELOVL2 was hypermethylated and downregulated in the samples from Tam resistance breast cancer patients compared with those from Tam sensitive patients. ELOVL2 was shown to recover Tam sensitivity up to 70% in the MCF-7/TamR cells and in a xenograft mouse model. ELOVL2 inhibits MCF-7/TamR cell proliferation and recovers Tam sensitivity (*Jeong et al., 2021*). In addition, after analyzing the overexpression data set GSE132614 of the gene ELOVL2, we found that the expression of GLI1 was up-regulated, and the expression of ECM-related gene set was changed (Fig. 5C) (*Jeong et al., 2021*). Some studies indicate that the organization of the extracellular matrix (ECM) can considerably alter the collective behavior of epithelial cells and increase the inclination for metastatic spread and invasive of tumors, suggesting that ECM organization can be a significant regulator of cell migration (*Egeblad, Rasch & Weaver, 2010*; *Levental et al., 2009*; *Lu, Weaver & Werb, 2012*). Andre et al. found that aligned ECM topography can trigger the EMT-like phenotype (*Park et al., 2019*). The above analysis indicates that gene ELOVL2 may affect tumor cell migration by changing ECM. These studies show that ELOVL2-AS1 may play an important role in the development, drug resistance, and immunotherapy of BRCA. Therefore, in-depth investigation of the reasons for the differences in the expression of ELOVL2-AS1 in BRCA subtypes, the specific mechanism of ELOVL2-AS1 in the process of tamoxifen resistance, and the role of ELOVL2-AS1 in immunotherapy are important research directions for the future.

## CONCLUSIONS

In conclusion, ELOVL2-AS1 is down-regulated in TNBC and might serve as a biomarker to distinguish the different subtypes of BRCA. It is also a potential prognostic marker for TNBC. The migration-suppressing effect of ELOVL2-AS1 might be mediated via regulation of key target genes involved in cilia or ELOVL2. In vivo or *in vitro* experiments are warranted to verify the underlying mechanism of ELOVL2-AS1 and its interactions with target genes in the future.

### Funding
The authors received no funding for this work.

### Competing Interests
The authors declare there are no competing interests.

### Author Contributions
- Mingda Zhu conceived and designed the experiments, performed the experiments, analyzed the data, prepared figures and/or tables, authored or reviewed drafts of the paper, and approved the final draft.

- Jingyang Zhang performed the experiments, authored or reviewed drafts of the paper, and approved the final draft.
- Guangyu Li analyzed the data, authored or reviewed drafts of the paper, and approved the final draft.
- Zhenzhen Liu conceived and designed the experiments, authored or reviewed drafts of the paper, and approved the final draft.

## Data Availability

The data is available at Figshare:

Zhu, Mingda (2022): eRNA analysis in breast cancer. figshare. Dataset. https://doi.org/10.6084/m9.figshare.17829620.v1.

Zhu, Mingda (2022): BRCA-ATAC and Gene expression analysis. figshare. Dataset. https://doi.org/10.6084/m9.figshare.17830154.v1

Zhu, Mingda (2022): Figure 13456. figshare. Dataset. https://doi.org/10.6084/m9.figshare.17830712.v1

## Supplemental Information

Supplemental information for this article can be found online at http://dx.doi.org/10.7717/peerj.13264#supplemental-information.

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
