# Peer review of "ELOVL2-AS1 inhibits migration of triple negative breast cancer"

_PeerJ, doi:10.7717/peerj.13264_

## Round 0.1 · original submission · Minor Revisions

Please provide additional information about sample size and repeats in each assay. It will be better to check the figure resolution before publishing.

·

Basic reporting

- I find this manuscript well-written and easy to follow. I would like to suggest raw data should be cited in the text. I saw you attached these files in the figshare platform, but you can add some text mentioning these links.

- Regarding figure organization, I think you can include the figure FC in the conventional numbering of the article.

Experimental design

- I think the design follows the hypothesis showing interesting results for proposing the ELOVL2-AS1 region for further analysis.

Validity of the findings

- Taking together the figures in the text and supplementary, the results seem valid to answer the hypothesis. No more comments.

Reviewer 2 ·

Basic reporting

The authors here report ELOVL2-ASI as a potential prognostic biomarker for TNBC. They identified key enhancer region in breast cancer by applying integrated analysis.
The overall paper is very well written with clear language.
The references are mentioned wherever is needed. However, it would be helpful to mention reference in line 217 to 220.
Figures, tables and raw data are well described.

Experimental design

The overall experiment is very well design. From data extraction, eRNA analysis to cell assays support the hypotheses. Controls are included wherever necessary. It would be interesting to know how many MDA-MB-231 cells were used for lentivirus infection.

Validity of the findings

The data provided by the authors is sufficient to support the hypotheses. The results are very well explained and clearly stated. The data presented is supported by sound statistics.

The conclusions are clearly stated and connected to the original research question.

Reviewer 3 ·

Basic reporting

This is an interesting manuscript that the authors have taken good use of several bioinformatics tools to identify a research target and explore its functions and possible regulatory mechanisms. This reviewer enjoyed reading it and would like to a few suggestions.

Experimental design

To better support that ELOVL2-AS1 could be used as a therapeutic and its regulatory relationship with ELOVL2, the authors may consider a ELOVL2-AS1 knockdown assay (if possible) see how that affects expression of ELOVL2 and BRCA cell migration.

Validity of the findings

The expression of ELOVL2-AS1 in the OE cell is dramatically different from that of the control, will this affect cell viability and physiology characters?

Additional comments

The authors should provide information on sample size and number of repeats in each assay in the method part as well as in figure legends and/or figures.
The resolution of figure 4, 5 (A) and (C) and figure S2 needs to be improved.

---

## Round 0.2 · accepted · Accept

Based on the reviewers' recommendations, I am delighted to inform you that your manuscript has been ACCEPTED.

·

Basic reporting

-

Experimental design

-

Validity of the findings

-

Additional comments

Dear authors,

I think the manuscript is well-written and easy to follow. Also, you have answered my minor concerns suffice. Therefore, I recommend this manuscript for its publication.